# Clonal Hematopoiesis and Mutations of Myeloproliferative Neoplasms

**DOI:** 10.3390/cancers12082100

**Published:** 2020-07-28

**Authors:** Lasse Kjær

**Affiliations:** Department of Hematology, Zealand University Hospital, Vestermarksvej 7–9, DK-4000 Roskilde, Denmark; laskj@regionsjaelland.dk

**Keywords:** myeloproliferative neoplasm, *JAK2*, clonal hematopoiesis, somatic mutations, mutation screening, cardiovascular disease

## Abstract

Myeloproliferative neoplasms (MPNs) are associated with the fewest number of mutations among known cancers. The mutations propelling these malignancies are phenotypic drivers providing an important implement for diagnosis, treatment response monitoring, and gaining insight into the disease biology. The phenotypic drivers of Philadelphia chromosome negative MPN include mutations in *JAK2*, *CALR*, and *MPL*. The most prevalent driver mutation *JAK2V617F* can cause disease entities such as essential thrombocythemia (ET) and polycythemia vera (PV). The divergent development is considered to be influenced by the acquisition order of the phenotypic driver mutation relative to other MPN-related mutations such as *TET2* and *DNMT3A.* Advances in molecular biology revealed emergence of clonal hematopoiesis (CH) to be inevitable with aging and associated with risk factors beyond the development of blood cancers. In addition to its well-established role in thrombosis, the *JAK2V617F* mutation is particularly connected to the risk of developing cardiovascular disease (CVD), a pertinent issue, as deep molecular screening has revealed the prevalence of the mutation to be much higher in the background population than previously anticipated. Recent findings suggest a profound under-diagnosis of MPNs, and considering the impact of CVD on society, this calls for early detection of phenotypic driver mutations and clinical intervention.

## 1. Introduction

Molecular diagnosis of myeloproliferative neoplasms (MPNs) have not only provided groundbreaking knowledge of their biology and genetic landscape but revolutionized the speed and accuracy of diagnosis. Identification of specific therapeutic targets through genetics has led to the introduction of novel therapies of these entities and very sensitive monitoring of treatment response. The earliest discovery of genetic aberrations associated with hematological malignancy was the (9;22) translocation or Philadelphia chromosome, resulting in a fusion between the genes coding for *BCR* and *ABL1*, identifying patients with chronic myelogenous leukemia (CML). The discovery of tyrosine kinase inhibitor therapies leading to molecular remission of the disease prompted the introduction of molecular assays measuring fusion transcripts to unprecedented sensitivity when monitoring treatment response [1]. The classical Philadelphia chromosome-negative MPNs include essential thrombocythemia (ET), polycythemia vera (PV), and primary myelofibrosis (PMF), which are characterized by driver mutations in a very limited number of phenotypic driver genes: *JAK2, CALR,* and *MPL,* involved in activation of the JAK-STAT signaling pathway [2]. In contrast, the hematological malignancies myelodysplastic syndrome (MDS) and acute myeloid leukemia (AML) display a wide variety of genetic abnormalities and no single predominant driver mutation has been identified [3,4]. The hematopoietic system is maintained by a pool of self-renewing multipotent hematopoietic stem cells (HSC) that produce a wide variety of highly polyclonal cell populations including granulocytes, monocytes, lymphocytes, and megakaryocytes [5]. The MPNs are considered to be initiated when changes in the genetic landscape of an HSC leads to clonal expansion and result in the generation of a malignant clone [6]. Although the presence of clonal hematopoiesis (CH) has been known for decades [7,8], the mutations associated with hematologic malignancies was expected to be an infrequent occurrence reflecting the rarity of the diseases. Initial investigations of the most prevalent MPN mutation—*JAK2V617F—*in the general population revealed it to be present in 0.1% of individuals and was strongly associated to the development of MPNs [9]. The following year MPN-associated mutations, such as *DNMT3A*, *TET2*, and *ASXL1*, were found in CH in healthy individuals [10,11,12] and, surprisingly, a study with increased sensitivity for detection of very small clones revealed that mutations in the genes for *DNMT3A* and *TET2* were nearly ubiquitous in individuals above the age of 50 [13]. In line with this, investigating the presence of the *JAK2V617F* mutation in the general population with an assay allowing the detection of very low frequencies revealed the prevalence to be more than 30 times higher than previously reported [14]. The discrepancy between the high prevalence of CH with driver mutations and the development of hematological malignancy may, in part, be a failure to diagnose patients or a stable pre-leukemic condition that requires further factors to progress. Required stimuli include, among others, additional mutations, elapsed time, life-style, and chronic inflammation. Bone marrow aging impacts the cellular composition and function of the hematopoietic system [15] and is accompanied by a sterile low-grade chronic inflammation [16] that has been hypothesized to be both an initiator and driver of MPNs [17]. Unfortunately, CH with a relatively low risk of progression to neoplasms is not a harmless condition but leads to decreased patient survival independent of hematological malignancies, increased risk of coronary heart disease, earlier onset of myocardial infarction, and is now accordingly considered a cardiovascular risk factor [11,18,19,20,21]. In this review I will mainly focus on Philadelphia chromosome negative MPNs and the MPN related ‘driver’ mutations considered to have impact on the disease. Accordingly, the subject of passenger mutations will not be addressed here. I will furthermore discuss the role of driver mutations for the detection of small hematopoietic clones that may lead to malignancy or cardiovascular disease (CVD). 

## 2. Mutations in Myeloproliferative Neoplasms

### 2.1. Discovery of Phenotypic Driver Mutations in Myeloproliferative Neoplasms

The first description of CML was published 175 years ago in 1841 and was the first cancer to be identified by a genetic aberration 115 years after its initial discovery. The Philadelphia chromosome was originally described as a small abnormal chromosome in the diseased cells but a decade later improved chromosome banding techniques identified it to be a reciprocal translocation between chromosomes 9 and 22 [22] providing both a diagnostic and a disease specific marker distinguishing CML from the Philadelphia negative neoplasms ET, PV, and PMF. In 2005 four laboratories made the breakthrough finding of the somatic *JAK2V617F* mutation (a G to T transversion in exon 14 of the *JAK2* gene) using a selection of genomic, functional, and genetic strategies [23,24,25,26]. The mutation in codon 617 of the gene led to a valine to phenylalanine amino acid change in the regulatory pseudo-kinase domain of the protein. Consequently, *JAK2* and its downstream JAK-STAT signaling is constitutively activated cytokine independently but also confers hypersensitivity to cytokine stimulation [23,27]. The mutation was subsequently found to be present in around 50% of ET and PMF patients and in approximately 97% of PV patients [28,29], whereas it was only occasionally identified in myelodysplastic syndrome (MDS), acute myeloid leukemia (AML), and refractory anemia with ringed sideroblasts with marked thrombocytosis (RARS-T) [30]. The following year the first gain-of-function mutation was discovered in the myeloproliferative leukemia virus oncogene (*MPL*) the gene coding for the thrombopoietin receptor resulting in constitutive activation of the receptor and downstream JAK-STAT signaling pathway [31]. Subsequently, during a screening for the newly found mutation, another was identified [32] and these two most recurring mutations W515L and W515K were found to be present in 5% of ET patients and up to 10% of PMF patients but absent in PV patients [33]. In 2007, close examination of patients with PV characteristics, but with no identifying *JAK2V617F* mutations, revealed novel mutations in a ‘hot-spot’ region of exon 12 in the *JAK2* gene [34]. Contrary to the point mutations *JAK2V617F* and *MPLW515*, a wide variety of *JAK2exon12* mutations were reported for the ‘hot-spot’ region. The most prevalent mutations were in-frame deletions of 3–12 nucleotides in addition to single nucleotide changes. However, in up to one-third of patients, complex mutations were seen. [35]. The *JAK2exon12* mutations are situated in a domain that acts as a linker between a src homology domain and the pseudo-kinase. Similar to the *JAK2V617F* mutation the *JAK2exon12* mutations result in cytokine independence and hypersensitivity [35]. A new era of mutation discovery dawned with the emergence of next generation sequencing (NGS) and whole exome sequencing of MPNs revealed somatic mutations in the gene coding for calreticulin (*CALR*) a chaperone involved in diverse functions such as the folding of newly synthesized glycoproteins and calcium homeostasis in the endoplasmatic reticulum (ER). At the time of the discovery, the NGS technology was challenged by short amplicons in addition to low coverage of the affected area in the *CALR* gene, which made identification of the large deletions in the gene difficult. Despite these obstacles two groups simultaneously published the identification of a large number of mutations in exon 9 of the gene including deletions and insertions, in addition to insertions and deletions (indels). Two mutations of the more than 50 reported variants—a 52 base pair deletion (type 1) and a 5 base pair insertion (type 2)—accounted for 80–90% of the mutations [36,37]. The *CALR* mutations were found in 60–80% of ET and PMF patients without *JAK2* and *MPL* mutations and could also be identified in rare cases of PV and RARS-T [38,39,40,41]. Despite the multitude of *CALR* mutations, they all shared the unique characteristics that they result in a +1 base-pair frameshift of the entire downstream coding sequence leading to a a collection of negatively charged amino acids being replaced with a positively charged chain [36,37]. The new amino acid sequence did not share homology with any known peptides but the wide variety of mutations resulting in an identical sequence suggested a gain of function and a high selective pressure for the feature. Further enquiries suggested that cellular transformation depended on the presence of a functional MPL receptor and it was discovered that the novel peptide in mutant CALR enabled the mutated protein to congregate in homomultimers. Acting like a cytokine this resulted in MPL receptor homo-dimerization, but without subsequently dissociation, leading to constitutive downstream activation of the JAK-STAT signaling pathway [42,43]. The *JAK2*, *CALR*, and *MPL* mutations thus converge on activation of JAK-STAT signaling, particularly through activation of the MPL receptor, and as they by themselves are sufficient to initiate and maintain the MPN phenotype in patients and murine models, they are considered phenotypic driver mutations (Figure 1) [44,45]. Compared to other cancers the hematopoietic neoplasms carry strikingly few mutations [46]—in MPNs, almost half of all patients bear only a phenotypic driver mutation and genetic lesions in younger patients below the age of 39 appear to be confined to the phenotypic driver mutations *JAK2*, *CALR*, or *MPL* [47,48]. However, additional somatic aberrations, identical to mutations found in MDS and AML, have been identified in >50% of MPNs. 

### 2.2. Additional MPN-Related Somatic Mutations

The other mutations identified in MPNs are not considered disease specific and they were only recently included in a diagnostic algorithm [48]. The most commonly identified changes are in genes involved in epigenetic regulation, spliceosome regulators, cytokine signaling pathways, and transcription factors, in addition to treatment related selection of clones with mutations in DNA damage response. The epigenetic regulators consist of two groups those involved in DNA methylation: *TET2*, *DNMT3A*, and *IDH1/2*, and two genes: *ASXL1* and *EZH2*, which are part of the polycomb repressor complex methylating histones to modulate chromatin structure [49]. 

#### 2.2.1. Epigenetic Regulators

*TET2* mutations are the most frequently encountered non-phenotypic driver mutations, observed in up to 22% of MPNs. Although the catalytic domain is most frequently affected, deletions, insertions, and substitutions have been observed in the entire sequence of the *TET2* gene. The resulting loss of function compromises its de-methylating ability, leading to DNA hyper-methylation. There is controversy regarding *TET2* mutations and whether they affect the risk of transformation to AML [47,50], but for *JAK2V617F* mutated MPNs, the sequence of mutation acquisition appears to have a clinical impact on the phenotype. Patients acquiring the *JAK2V617F* mutation prior to a *TET2* mutation were found to present with a more severe phenotype, were more likely to be diagnosed with PV compared to ET, and suffered from an increased risk of thrombosis [51]. Mutations in *DNMT3A* are found in less than 10% of MPNs and are found as nonsense/frameshift mutations across the entire coding sequence and especially a missense mutation in the methyltransferase domain (R882) leading to reduced enzyme activity and de novo DNA methylation [52,53]. The mutations are more prevalent in later stages of MPNs and similar to mutations in *TET2* the acquisition order of the mutation relative to *JAK2V617F* appears to influence the phenotype such that patients with initial *JAK2V617F* mutation have a tendency to present with PV or MF rather than ET [54]. IDH1 and -2 alterations in the catalytic site results interference with histone demethylation. The missense mutations have low frequency in patients with ET and PV but are found in up to one third MPNs that have progressed to AML and are suspected to be involved in transformation to the acute phase. [55,56]. *ASXL1* mutations are relatively uncommon in ET and PV, but they become frequent in PMF and AML where they can be found in up to 30% and 50% of the patients, respectively. The frameshift and nonsense mutations usually target exon 12 of the gene but can be found throughout the coding sequence and result in loss of the c-terminus. ASXL1 associates with a range of chromatin-altering proteins and although the functional consequences of the mutations are unclear it is evident that they impart a poor prognosis on PV and PMF patients [56,57,58,59]. The entire coding sequence of *EZH2* can harbor amino acid altering or truncating mutations reducing, methylation of H3K27 and may enhance the function of HSCs [60]. The prevalence of the mutations increases from a few percent in ET and PV up to around 10% in PMF and around 15% in post-MPN AML and are an indication of a poor prognosis [47,61]. 

#### 2.2.2. Spliceosome Regulators

The most frequently observed mutations in spliceosome regulators include the genes *SF3B1*, *SRSF2*, *U2AF1*, and *ZRSR2*. *SRSF2* and *U2AF1* belong to a group of factors that are involved in the regulation of mRNA splicing and contain the most prevalent of the spliceosome regulator mutations. The mutations are associated with a poor prognosis and the prevalence in PMF patients of *SRSF2* and *U2AF1* mutations is 15% and 22%, respectively [62,63,64,65]. Mutations in the *ZRSR2* gene are predominantly non-sense and frameshift mutations that accumulate during the progression of disease and are generally found in PMFs [48]. The *SF3B1* mutations, mainly found in exons 12–16, were initially associated with MDS but are in the context of MPNs usually found in patients with PMF or in an MPN/MDS phenotype with RARS-T characterized by co-occurrence of the *JAK2V617F* mutation [66,67]. 

#### 2.2.3. Regulators of Cell Signaling, Transcription Factors, and DNA Damage Response

Mutations in regulators of cell signaling are rare in MPNs and include alterations in *SH2B3 (LNK)*, *CBL*, *NRAS/KRAS*, and *PTPN11*. The known clinical consequences of these alterations are that *SH2B3* mutations are associated with shortened survival in ET patients, *NRAS/KRAS* associates with leukemic transformation and *PTPN11* with shortened survival [68,69,70]. *RUNX1* and *NFE2* mutations are relatively rarely seen in MPNs but *RUNX1* alterations increase in prevalence with progression to post-MPN AML and is linked to decreased survival [47,68,69,70,71]. The final group of gene mutations are not normally detected in the early phases of disease, but are commonly associated with progression of disease (*TP53)* or in most cases appear to be treatment related *(PPM1D)* [47,48,72,73]. 

### 2.3. Algorithm Integrating NGS Data in Patient Prognosis

No longer in its infancy, NGS used for routine analysis of patient samples generate a wealth of data that might be overwhelming for clinicians striving to predict patient’s outcomes. The impact of mutations in specific genes are gradually being unraveled, but patient’s outcomes cannot be predicted by mutational status alone and requires assimilation of clinical values. To address the challenge of stratifying the many-featured MPN patients by compounding clinical and laboratory insights, 69 myeloid cancer genes in over 2000 patients were recently analyzed. Eight mutational subgroups were identified, where *TP53* mutations imposed a particularly dismal prognosis with risk of progression to AML. Mutations in genes of the RNA spliceosome or chromatin regulation were typically in patients with PMF. *MPL* mutations were associated with an elevated risk for transformation to AML and homozygosity for *JAK2V617F* usually found to correlate with PV diagnosis. Patients with no detectable mutations, in general young female ET patients, had especially benign outcomes [48]. The algorithm requires further validation in MPN patients as the study was skewed toward including ETs, but nevertheless remains an important initial step in providing a user-friendly tool for data integration and personalized medicine supporting clinicians in difficult decisions concerning the diagnosis and treatment of MPNs.

## 3. Hematopoietic Stem Cell Clones with Phenotypic Driver Mutations

### 3.1. One Mutation Several Phenotypes

Several lines of evidence suggest that the driver mutations affect HSCs and result in clonal expansion of differentiated progeny, but the question remain when they occur and the factors involved in the development of clonal dominance. The selective advantage in either survival or growth appears to be dependent on other features such as additional mutations or environmental factors. Gradual acquisition of additional mutation may form a rate limiting step for the development of hematologic malignancies, but as they are among the cancers presenting with the lowest number of mutations, this suggests a certain resilience of HSCs [46]. The phenotypic mutations, such as *JAK2V617F* and *CALR* mutations, confer a strong selective advantage allowing a single mutation to continuously increase [47,48,74,75]. However, for the *JAK2V617F* mutation, it is still a mystery how a single mutation can give rise to different disease phenotypes (ET, PV, and PMF). The phenotypes appear to be associated with different tumor loads [76,77] where ET is characterized by heterozygosity for the mutation and PV by homozygosity [78], suggesting they represent different disease entities or a biological continuum from ET over PV to PMF (Figure 2A). During development of the disease it was observed that some heterozygous patients lost the wild type allele, becoming homozygous in a process known as mitotic recombination resulting in uniparental disomy [25]. Surprisingly, homozygosity appeared to arise multiple times independently in *JAK2V617F* mutated clones in both ET and PV patients. The difference appeared to be that in PV patients a single homozygous sub-clone expanded and became the dominant clone, whereas it remained a sub-clone in ET patients [79]. An important clue to this emerged with the discovery that not only did the type of mutation have an impact but also the order of occurrence as previously mentioned. Patients acquiring *TET2* mutations presented with a more severe phenotype resembling PV if the clone already harbored the *JAK2V617F* mutation (the ‘*JAK2*-first’), whereas patients with ‘*TET2-*first’ clones appeared to have a less aggressive ET-like phenotype. The *TET2* mutations arose independently of hetero- and homozygosity but ‘*JAK2-* first’ had a striking expansion of the homozygous clone [51]. This may be a more widespread phenomenon, as the order of mutations acquisition also affects the phenotype for *DNMT3A* and *JAK2V617F* double mutants; however, whereas the ‘*TET2-*first’ patients were older at presentation, no age difference were found in the *DNMT3A* groups, possibly because of a smaller number of patients [54]. This raises the relevant issue with the presence of mutations and their order for individuals in the ‘low-allele burden’ risk group and could help in predicting which are early pre-MPNs and which, if any, may be a more benign CH (Figure 2). 

### 3.2. Phenotypic Driver Mutations in Lymphoid Cells

A key difference between *JAK2V617F* and *CALR* mutated patients is that homozygosity is rarely detected in the latter [80]. Initially, it was suggested that *JAK2V617F* mutated ET was more likely to have polyclonal hematopoiesis than *CALR* mutated [81], but this was subsequently disputed by another group claiming that the *HUMARA* assay used in the previous study did not always reflect the X-chromosome inactivation accurately and that *CALR* had a weaker suppressive effect on normal myelopoiesis than *JAK2V617F,* supportive of the decreased severity [82]. *CALR* mutations are considered an early event [36,37] and *CALR* rarely presents with a low mutant allele burden, not only for diagnostic samples [67] but also in the background population [14], suggesting that, compared to *JAK2V617F* mutants, *CALR* mutated cells relatively quickly out-competes wild type cells (Figure 2A). Interestingly, although phenotypic driver mutations induce myeloid differentiation bias, the *JAK2V617F* and *CALR* mutations are detectable in lymphoid cells in a subset of patient applying both to B- and T-cells where patients had mutations in one or both cell types. Detection of *JAK2V617F* mutations in lymphocytes was associated with an overall higher mutant allele burden types and prolonged disease duration. It should be noted that almost all patients in the *JAK2V617F* study were PV patients with many homozygous carriers, whereas the *CALR* study mainly included heterozygous pre-PMF and PMF patients [83,84]. For *JAK2V617F* positive patients with very high allele burdens >90% and *CALR* mutated patients with heterozygous allele burdens close to 50%, it is expected that the phenotypic driver mutations are present in virtually all HSCs in the bone marrow. Consequently, the majority of B and T cells would be expected to carry the mutation but this was not observed as the mutant allele burdens were generally lower in lymphocytes than in granulocytes. This is a surprising finding to be interpreted in several ways. First, the HSCs carrying the mutation have a very limited proliferative advantage, if any, so that the lymphopoiesis remains polyclonal, whereas slightly more differentiated progenitors are affected by the mutations and dominate these populations, but with a myeloid bias. Second, the mutations differentially target multi-lineage progenitors and the specific differentiation potential of the affected progenitor influences whether lymphoid cells will carry the mutation are observed. These interpretations demand that the mutations confer a much stronger proliferation potential to the myeloid lineage and that the gap between the clone durability of the multi-lineage progenitor and HSC is smaller than hitherto appreciated, which recent studies have indicated [5]. Third, it may be influenced by the longevity of B and T cells, and although it appears that in particular the *JAK2V617F* clones develop over decades, the B- and T cells may represent cells that were matured prior to (or during) expansion of the mutated HSCs consistent with the prolonged disease duration associated with the appearance of the mutations in lymphocytes. 

### 3.3. Order of Mutation Acquisition as an Indicator of when in Life the Phenotypic Driver Mutation Occurred

Tracking mutations in exomes of hematopoietic progenitor cells identified that most mutations found in AML genomes actually occur before the initiating event and increase linearly with age, presenting a ‘molecular clock’ for the age of the affected individual [85,86]. As suggested earlier, the *JAK2V617F* clone can develop at variable rates ranging from years to decades [75,87], and in a small fraction of patients, the low allele burdens appear to remain stable for extended time periods, suggesting a very slow expansion in some individuals [88]. Since younger individuals with the *JAK2V617F* mutation usually only present with this mutation, it is possible that the *TET2/JAK2V617F* or *DNMT3A/JAK2V617F* acquisition order is a temporal issue, where the inevitable mutational accumulation of *TET2/DNMT3A* is an indicator of the time point in the individual’s life, when the *JAK2V617F* mutation occurs. Thus far, it is unclear why the homozygous clones in ET patients do not expand to dominance, but several elements could be influential, including intrinsic factors such as additional age-related mutations (*TET2*) and extrinsic factors such as systemic inflammation or inflammation in the complex bone marrow environment. An early occurrence of *JAK2V617F* before the appearance of *TET2/DNMT3A* provides time for uniparental disomy to develop; alternatively, the more aggressive PV phenotype may be caused by an earlier stage HSC/progenitor that acquired the genetic lesion. The *JAK2V617F* mutation has been considered to induce genomic instability [89], and although it has been argued that these results were obtained from model systems that might not reflect human MPNs [90], it is intriguing to consider if the increased genetic instability could be a consequence of cellular hyper-proliferation/turnover accelerating the ‘molecular clock’—a premature aging, increasing the risk of acquiring additional mutations and subsequent malignant transformation. 

## 4. Clonal Hematopoiesis and MPN-Related Mutations in Healthy Individuals

During healthy hematopoiesis, HSCs give rise to highly polyclonal cell populations. The emergence of clonal dominance suggests the acquisition of a survival or cell growth advantage. The functional consequences of CH are largely unknown, but the main concern is that continuous expansion of a single clone with one or more traits de-regulating the mechanisms controlling intended cell growth and death generates an ever-increasing pool of cells poised on the verge of transformation. Emergence of a malignancy is considered to be initiated by gradual accumulation of genetic insults in healthy cells that result in excessive clonal expansion and subsequent alterations that ultimately leads to full transformation [91]. However, blood cancers span a wide spectrum of malignancies from chronic MPNs with astonishingly few known mutations to aggressive AML that often displays complex mutational patterns [3,4]. As previously mentioned, the disparity between development of CH and blood cancer emphasize the importance of uncovering which risk factors hold clinical relevance for persons with CH both for the development of malignancies and, as will be addressed later, CVD.

### 4.1. First Discoveries of Clonal Hematopoiesis in Healthy Individuals

X-chromosome inactivation studies of the 1960s laid the foundations for clonality research where mosaicism in the paternal and maternal x-linked genes was demonstrated in females and used for the seminal description of the single-cell-of-origin in a tumor [92,93,94]. A decade later, this was used to unveil the clonal origin of granulocytes, red blood cells, and platelets in PV [95]. In the 1990s, asymmetrical inactivation was detected in healthy women, but this observation was initially considered to be caused by the variety of assays used and limited population sizes [96,97]. However, subsequent studies demonstrated inter-tissue asymmetry in X-chromosome inactivation showing particularly high skewing in blood cells [8,98]. Importantly, these skewed patterns were more frequently found in older women than in infants, suggesting either an age-related clonal loss or an accumulation of a hematopoietic clone [7]. Not until 2012 did exome sequencing reveal the skewing to be associated with mutations in a gene that was related to myeloid cancers. Somatic mutations were identified in the *TET2* gene of nearly 6% individuals with skewed X-chromosome inactivation, but otherwise harbored normal hematologic values. Although the skewing was considered to be related to clonal expansion, only a small fraction of the mosaicism was thought to be related to mutations in the identified driver gene [99]. Three similar studies demonstrated age-dependent clonal mosaicism, indicating clonal hematopoietic expansion and added to the findings by linking this to an increased risk for development of hematologic cancers [100,101,102].

### 4.2. Mutations as Molecular Markers of Clonal Hematopoiesis

Introduction of NGS has enabled a tremendous upscaling of discovering genomic variants. In 2014, three studies sequencing exomes in large cohorts, reported age-related mutations in driver genes associated with myeloid neoplasms. The first study analyzed blood samples used as germline controls for solid tumors for The Cancer Genome Atlas [10] and two screened for germline risk-factors in type-2 diabetes [11] and in schizophrenia [12]. The recurrent mutations included alterations in *DNMT3A*, *ASXL1*, *TET2*, *JAK2*, and *SF3B1* and the two studies not associated with cancer revealed an increased risk of developing a hematological malignancy [11,12]. Although the increased risk for myeloid cancers might be unsurprising in individuals with CH, the data also indicated an unsuspected increased risk of CVD, which was subsequently confirmed [19]. These findings suggested the discovery of a pre-neoplastic stage and obscured the distinction between clinical entities. The observations were termed CH of indeterminate potential (CHIP) or age-related CH (ARCH). To narrow the definition of these pre-clinical entities, the CHIP definition was determined to require absence of previous hematological malignancy and a mutant allele burden of least 2% or variant allele frequency (VAF) of 0.02 [103]. Astonishingly, the following year, a study using an error corrected targeted approach enabled a sensitivity of 0.03% demonstrated *TET2* and *DNMT3A* mutations to be virtually all-pervasive after the age of 50 in healthy individuals [13]. The low prevalence of myeloid malignancies in persons with MPN-related mutations high-light the need for introducing clinically relevant cut-off values reflecting the biological aspects of CH and integrating information on clone size, identity of mutations, and order of acquisition to aid in estimating the risk of developing a malignancy, subsequent prognosis, and treatment decisions. However, since the phenotypic driver mutations such as *JAK2V617F* and *CALR* are sufficient to initiate MPNs, it is a conundrum that they can be detected in apparently healthy individuals. 

### 4.3. CH Does Not Appear to Involve Genetic Predisposition 

It has been suggested that the genetic background with the 46/1 haplotype might form a predisposition for the development of the *JAK2V617F* phenotype [104,105], and other studies have found that an 8-base pair deletion in *TERT* [106] was associated with the presence of CH and that *TET2*, but not *DNMT3A,* mutations exhibited familial aggregation [107]. These studies did not investigate if the predisposition was genetic or environmental; however, this was recently addressed by two studies investigating 299 elderly twin pairs with 20 years follow-up [108] and 79 twin pairs with 4–5-year follow-up of clonal trajectories [109], and neither study detected genetic predisposition for mutations resulting in CH.

## 5. Phenotypic Driver Mutations in Background Population

Studies investigating the prevalence of *JAK2V617F* in larger cohorts unselected for blood disorders have reported values from 0.18% [11] and 0.19% [12] to 0.61% [75]. To address the prognosis in *JAK2V617F* positive individuals without MPNs the ‘Copenhagen General Population Study’ investigated almost 50,000 individuals from the general population. Using a sensitive qPCR technique, they found 0.1% *JAK2V617F* positives, where 48 of 63 individuals eventually developed MPNs [9,87] and at follow-up it was observed that the mutant allele burden increased 0.55% per year. Although it was based on a limited number of patients, the authors proposed a 2% cut-off value for *JAK2V617F* for disease vs. no disease mirroring the mutant allele burden in the CHIP definition by Steensma et al. [103]. As concluding remarks, the authors stressed that medical attention was advised for individuals below this threshold, as the results suggested that they were likely to develop MPNs [87]. 

### 5.1. Quantitating Phenotypic Driver Mutations

The detection of *JAK2V617F* was, at the time of discovery, achieved by Sanger sequencing, but it rapidly became clear that as a result of the suboptimal sensitivity, many patients carrying the mutation eluded detection. Introduction of allele-specific PCR enhanced the detection rate by improving the sensitivity of 10–20% to approximately 3% mutated alleles, but still only provided a qualitative measurement [24]. Quantitative determination by qPCR was predominantly developed for measuring the tumor load, but also for the superior sensitivity, enabling the detection of low mutant allele burdens. The *JAK2V617F* mutant allele burden holds value in prognosis, as it appears to be associated with the MPN phenotype, the risk for developing thrombosis in both ET and PV patients, and blood cell counts (BCC), and predicts the development of fibrosis [76,77,110,111]. Furthermore, in PMF, it may be an early warning of progression to AML or bone marrow failure as a mutant allele burden below 25% is linked to reduced survival [112]. Monitoring tumor load has been crucial for evaluating the effect of therapies that induce molecular remissions by targeting the malignant clone such as interferon-alpha-2a (IFN) [113]. The initial mutant allele burden measurements in *CALR* patients were performed using pseudo-quantitation by fragment analysis and it was observed that, similar to *JAK2V617F,* the mutant allele burden appeared to be increased in advanced stages [40]. A subset of *CALR* patients also demonstrated a reduction of the mutant clone with IFN treatment, although it appeared less efficient in these patients compared to carriers of the *JAK2V617F* mutation [114,115,116]. However, the sensitivity of fragment analysis is suboptimal for monitoring molecular remission, and qPCR analysis exposed a bias in the quantitating type 1 mutations when using fragment analysis. For longitudinal patient samples qPCR furthermore indicated that the dynamics of BCC and mutant allele burdens closely resembled each other during IFN treatment [117]. 

Designing very sensitive qPCR assays for quantifying the mutant allele burdens of *JAK2V617F* and *CALR* has been challenged by cross-reactivity of the oligonucleotides specific for the mutant sequence with the wildtype sequence, particularly for the single nucleotide polymorphism of *JAK2V617F* and the *CALR* mutational hot-spot that contains many repetitive sequences. Several qPCR assays have been developed for the quantitation of *JAK2V617F* and *CALR,* and it was established that the assays were superior when specificity was based on the primers [118,119], which could be further improved by introducing a 3′ mismatch intentionally destabilizing primer binding [84,117]. It is an advantage that qPCR can analyze a broad dynamic range, but as the quantification is dependent on setting a certain threshold for determining which amplification cycle should represent the quantitative value, factors such as primer quality, reagents, thermocycler, and lab technician can have a huge impact on the accuracy of the quantification. Furthermore, performing PCR analysis in a tube with very few mutated alleles in a background of high copy numbers of competing wild type sequences remains highly challenging. Digital PCR (dPCR) for detection of rare mutant variants with very low copy number in a high wild type background was given name in a study of the *ras* oncogene [120]. Several iterations of the technique has been developed and the topic has been extensively reviewed in recent years [121,122], including its applications in hematological malignancies [123,124]. A variation of the technique provided by Bio-rad using miniscule 1 nL droplets in an oil emulsion as reaction chambers is termed droplet dPCR (ddPCR). The small size of the reaction chambers only allows for a few copies of the target to be present in every reaction and practically abolishes the issues with competitive primer binding seen for qPCR. Around 20,000 individual reaction chambers are generated for each well, allowing Poisson statistics to determine the absolute copy number of the target minimizing run to run variation and increasing accuracy. The two-color technology allows for the determination of the mutant and wild type allele, concomitantly removing pipetting errors, which further represses any unspecific binding, and allows each well to act as its own loading control. 

### 5.2. Deep Molecular Screening for Phenotypic Driver Mutations in the Background Population

We performed a cross sectional study using ddPCR to investigate the prevalence of very low mutant allele burdens of the phenotypic drivers *JAK2V617F* and *CALR* in 19,958 individuals from the cross-sectional Danish General Suburban Population Study (GESUS) [14]. We included *CALR* type 1 and type 2 in the screening, as these mutations comprise 80–90% of observed *CALR* cases and the third most common *CALR* mutation (type 3) only account for 1.7% [36,37]. Because of the low expected prevalence of the mutations and the considerable number of individuals, we pooled four samples, achieving a level of detection of 0.009% for *JAK2V617F* and 0.01% for *CALR*, where a positive well prompted reanalysis and quantification of each individual sample reducing the reagent cost per sample to below 7€. The cross-sectional design of the study meant 16 patients already diagnosed with MPNs were a part of the cohort. We identified 613 *JAK2V617F* positive individuals and 32 *CALR* positive individuals, corresponding to a 3.1% and 0.16% prevalence, respectively [14]. The *JAK2V617F* allele burden was significantly lower than that of *CALR* and the expected ratio of *JAK2V617F:CALR* of 5.6:1 [37,125] was skewed to 19:1. *CALR* positives were more than three times as likely to have a MPN-diagnosis in agreement with the hypothesis that *CALR* mutated cells develop more rapidly into MPNs [126]. Consistent with the lower assay sensitivity of previous studies analyzing for *JAK2V617F* reporting a lower prevalence, the majority of *JAK2V617F* positives in our study presented with a mutated allele burden below 1%. This is consistent with a very long pre-clinical phase from first hit to diagnosis [74] but, importantly, those below 1% mutated alleles were frequently associated with a distinct MPN biochemical profile implying 42% of mutated non-MPNs to have elevated BCC [14] (Figure 3). Importantly, this observation suggests a substantial degree of under-diagnosis in the background population. Early detection and treatment of pre-MPNs with a low mutant allele burden and concurrent elevated BCC is crucial as undiagnosed MPNs are in risk of experiencing repeated debilitating thrombotic events 5–10 years prior to diagnosis [127,128]. For low allelic burdens of phenotypic driver mutations, the CHIP-term thus may be misleading as they may represent a pre-MPN in the ‘latent form’ of myeloproliferative disease that recent research suggests is no longer of indeterminate potential and should receive additional medical attention as advised by Nielsen et al. [87].

## 6. MPN-Related Mutations in Cardiovascular Disease 

Aging is not only associated with the development of CH, but is the best explanatory variable for the risk of developing CVD [129]. In addition to the classic cardiovascular risk factors such as obesity, smoking, and hypercholesterolemia [130], CVD is also promoted by age-related sterile inflammation similar to MPNs [16,131,132]. Although it was uncovered that loss of function mutations in *TET2* induce pro-inflammatory processes [133,134], it nevertheless came as a surprise when NGS analysis of large cohorts discovered that *ASXL1*, *TET2,* and *DNMT3A* mutations doubled the risk for CVD, whereas the *JAK2V617F* mutation itself increased the risk 10-fold [11,19]. In line with this, patients carrying *TET2* and *DNMT3A* mutations had an increased mortality linked to complications of heart failure as the underlying cause and this was strongly coupled to the size of the mutated clone indicating a causative role in the pathogenesis [135]. Aging may thus be a proxy for the risk of acquiring somatic mutations and subsequent development of CVD. 

### Mutated Myeloid Cells Play Essential Role in Atheroclerosis and CVD

Atherosclerosis is the chronic inflammatory disease that forms the basis of the majority of CVDs. The process is initiated when monocytes, in response to a cytokine stimulus, bind to the endothelial lining of the arteries, infiltrate the vessel wall, proliferate, and differentiate into macrophages [136]. The macrophages engulf lipids and transform into foam cells, a key component in plaque formation [137]. Elevated numbers of monocytes are accordingly associated with an increase in plaque formation and acceleration of atherosclerosis [138]. Subsequent to the formation of the plaque, macrophages may compromise the integrity of the fibrous shell that covers the interior by protease release, such as cathepsin K and matrix metalloproteases leading to plaque rupture, leading to potential myocardial infarction or stroke. Another type of myeloid cells, granulocytes, contribute to the pathogenesis by accumulating at rupture sites and form occluding nuclear extracellular traps (NET) that aggravate thrombus formation [139]. The risk for venous and arterial thrombosis correlates with leucocyte numbers and for individuals carrying the *JAK2V617F* mutation, this translates into a considerable increased risk for thrombosis [140,141]. Furthermore, the *JAK2V617F* mutation appears to amplify the tendency of granulocytes to generate NET, and thus acts on several cellular levels increasing the risk of thrombus formation [142]. 

## 7. Conclusions 

The MPNs are propelled by phenotypic driver mutations revolving around the activation of the JAK-STAT signaling pathway. The gene dosage of these mutations is strongly associated with the disease subtype, but it is thus far unclear why the homozygous clones that are present in ET patients do not outcompete the heterozygous clones as seen in PV patients. Involvement of additional somatic MPN-related mutations, especially in genes coding for epigenetic regulators and the spliceosome, and surprisingly, the order of mutation acquisition, affects the developmental trajectory of the MPN phenotype and subsequent progression. The mechanisms governing how this affects the phenotype remains unclear, but may involve the identity of the cell initially acquiring the mutation or a potential conferred to the cell by the mutation itself. It has become clear that the acquisition of MPN-related somatic mutations and subsequent development of CH is inevitable with aging, where they can be considered as a form of ‘molecular clock,’ describing the number of cell divisions and risk of stochastically obtained mistakes at critical positions. Young MPN patients rarely present with other mutations than the phenotypic driver mutations, suggesting that the other MPN-related mutations occur later in life. The order of acquisition might be a ‘timer’ of when in life the phenotypic driver was acquired and the more severe *JAK2*-first phenotype could be an indication that it was an instance where the patient acquired the phenotypic driver early in life. The growth rate of clones with phenotypic drivers is relatively slow and it may take decades for a malignant clone to expand sufficiently to cause clinical symptoms dependent on additional factors affecting the speed of growth, including, among others, life-style, inflammation, and the type of driver mutation. It is becoming evident that the clear distinction between HSC and long-lived progenitor is becoming increasingly diffuse. The phenotypic drivers skew differentiation towards the myeloid cell lineages, but for a subgroup of patients, there is involvement of lymphoid cells with the phenotypic drivers, and these are typically associated with prolonged disease duration. It is currently unknown how the order of mutations affects the cellular or environmental context and results in a more severe phenotype. It will be interesting to investigate how much the phenotype and severity of disease is linked to the MPN-related mutations—what is the impact of alteration in epigenetic regulators and spliceosomal machinery, and does the *JAK2* mutation render the cell particularly vulnerable to the impact of subsequent mutations? Is the difference in phenotypes linked to the inherent potential of the targeted cell so that it reflects where, in a range of very long-lived progenitors with varying potential, the mutation occurs? Is it dependent on the age-related fitness of the targeted cell so that the pathogenesis is affected by when in life the phenotypic driver mutation is acquired? Clonal hematopoiesis emerging with age is likely to represent multiple different entities, developmentally influenced by several factors such as inflammation, concomitant mutations, and the type of mutation, but also by a temporal component, both with regard to time-dependent accumulation of additional genetic insults and size of the clone. The surprisingly large number of individuals in the background population carrying phenotypic driver mutations with concomitant elevation of BCC suggests a substantial under-diagnosis of MPN in the background population. The MPN diagnosis is in many cases founded on a long history of thrombosis in the patient, leading to debilitating complications, and as the first symptom in a large subset of patients with thrombosis is sudden death, an underlying MPN will consequently never be discovered, concealing a higher actual prevalence of the malignancies [143]. The low number of individuals with phenotypic driver mutation-associated CH that eventually receive an MPN diagnosis is initially comforting, but may on the other hand conceal a disturbing reality of massive under-diagnosis. Furthermore, the MPN related mutations, especially the phenotypic driver *JAK2V617F* is associated with an increased risk of CVD. As CVD present the primary cause of death globally, the high prevalence of driver mutations in the aging background population raises concerns beyond the development of hematologic cancers and the possibility that an underlying CH or occult MPN drives the pathogenesis strongly advocates for early detection and intervention of these (pre-) clinical entities. 

## Figures and Tables

**Figure 1 cancers-12-02100-f001:**
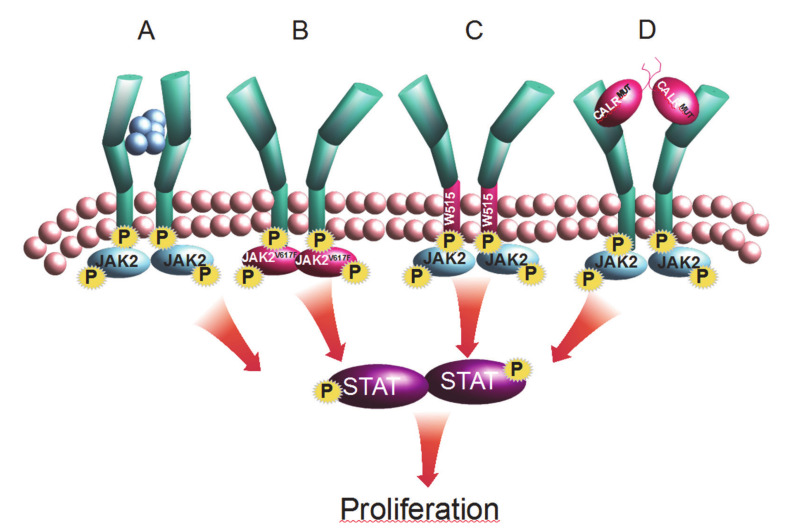
The phenotypic driver mutations of myeloproliferative neoplasms (MPNs) act through JAK-STAT signaling. JAK2 mediates downstream signaling for type 1 cytokine receptors such as the granulocyte colony stimulating factor receptor (G-CSFR), the erythropoietin receptor (EpoR), and, as exemplified in this figure, the thrombopoietin receptor (MPL). (**A**) In the normal cell ligand binding leads to phosphorylation of JAK2 and the receptor resulting in downstream phosphorylation of STAT proteins subsequently translocating to the nucleus where transcription of genes involved in cell proliferation and survival is initiated. (**B**) Mutated JAK2 leads to constitutively activated downstream signaling in the absence of ligand and enhanced signaling in the presence of ligand. (**C**) The MPL mutations affecting peptide W515, similar to mutations in the JAK2 protein, result in constitutively activated downstream signaling. (**D**) The novel peptide of the CALR protein, resulting from the +1-frame shift, introduces a new functionality for the CALR protein that leads to CALR-induced homo-dimerization of the MPL receptor and constitutive activation of the receptor. Proteins with mutations are depicted in red; phosphorylation is indicated with a P on yellow background.

**Figure 2 cancers-12-02100-f002:**
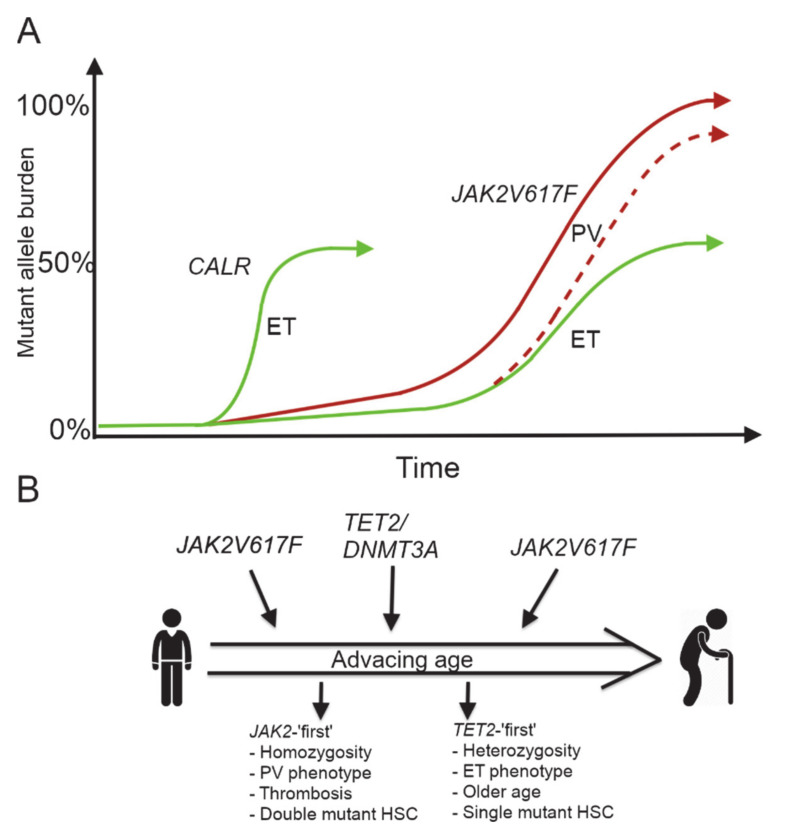
Expansion of the MPN clone is affected by both the type of phenotypic driver and co-mutation acquisition order. (**A**) The *CALR* allele burden is considered to expand rapidly but homozygosity is rare and is virtually always associated with the essential thrombocythemia (ET) phenotype (green curve to the left). Different disease trajectories are seen for patients carrying the *JAK2V617F* mutation. Homozygosity for the mutation arises frequently in patients, but a homozygous clone is more likely to become the dominant in patients diagnosed with polycythemia vera (PV) (red curve). The ET trajectory for patients carrying the *JAK2V617F* (green curve to the right) will occasionally progress into a PV phenotype indicating a ‘biological continuum’ from ET to PV (dashed red line). (**B**) The acquisition order of *JAK2V617F* relative to *TET2* and *DNMT3A* influences the disease phenotype. If the *JAK2V617F* occur prior to *TET2* this results in a more severe phenotype, with the homozygous *JAK2V617F/TET2* clone becoming dominant, whereas *TET2* first leads to a predominantly heterozygous phenotype, where the *JAK2V617F* clone is reluctant to outcompete the *TET2* mutated clone.

**Figure 3 cancers-12-02100-f003:**
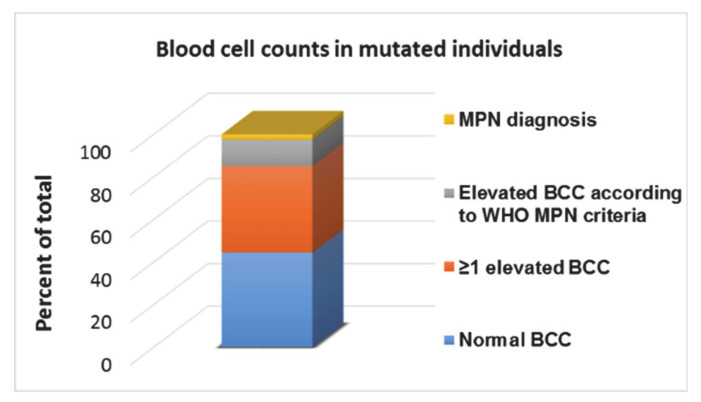
Elevated blood cell counts (BCC) in *JAK2V617F* and *CALR* mutation positives in the general population. Deep molecular screening on 19,958 individuals of the 21,205 total from the Danish General Suburban Population Study (GESUS) cohort, revealing 645 to be positive for *JAK2V617F* (N = 613) or *CALR* type 1 and type 2 (N = 32). Normal BCC was found in 44% (N = 287), 41% (N = 262) had elevated BCC including hematocrit, erythrocytes, leukocytes, and platelets, whereas 12% (N = 80) had elevated BCC counts fulfilling WHO MPN criteria and 2.5% (N = 16) were already known patients with an MPN diagnosis.

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
