# Peer review of "Clonal Hematopoiesis and Mutations of Myeloproliferative Neoplasms"

_cancers, 2020, doi:10.3390/cancers12082100_

Round 1

Reviewer 1 Report

In this review the Authors mainly focus on Philadelphia chromosome negative MPNs, their driver mutations and detection of small hematopoietic clones that may lead to malignancy or cardiovascular disease (CVD).

Minors Revision

- In line 73 page.4. - a G to T transversion in exon 14 of the JAK2 gene – should be put between parenthesis.

- In lines 87-89, page 4 The phrase  “Contrary to the point mutations reported for  JAK2V617F and MPL a wide variety of mutations were reported where the most prevalent were in-frame deletions of 3-12 nucleotides and single nucleotide changes but up 33% of patients harbored complex mutations” is confusing. It must be clarified.

- In line 92 page 4 the dash should be erased.

-In line 103 page 5 a  reference should be add  as recently Quattocchi et al reported the identification of no canonical mutations affecting CALR 3’-UTR in JAK2 mutation-negative patients with MPN diseases resembling PV (Quattrocchi A, Maiorca C, Billi M, Tomassini S, De Marinis E, Cenfra N, Equitani F, Gentile M, Ceccherelli A, Banella C, Mecarocci S, et al. Genetic Lesions Disrupting Calreticulin 3'-Untranslated Region In Jak2 Mutation-Negative Polycythemia Vera. Am J Hematol. 2020 Jun 22. doi: 10.1002/ajh.25911.

-In line 122 page 6 the dash should be erased.

- Primary  myelofibrosis in line 176 page 8 is indicated PMFs in the rest of the paper is write PMF. It should be the same way in all the manuscript.

- In line 181 page 8 “alterations are that SH3B2” must be change with “ alterations are that SH2B3”

- In line 226 page 10 and 349 page 15 the dash should be erased.

Reviewer 2 Report

Dr. Lasse Kjær provided a comprehensive understanding of the mutations in CH and MPN, the manuscript was organized well, and the content is abundant. There are some minor issues, here are my points:

  1. Periods were missed in some sentences, like Line 91, Line 105, and so on, please revise the manuscript.
  2. Line 66, “2.1. Discovery……”, Line 133, “2.1. Additional……”, Line 133 should be “2.2.”, so the rest of this paragraph needs corrections.
  3. Line 136 introduced about five categories, including epigenetic regulation, spliceosome regulators, cytokine signaling pathways, transcription factors, and mutations related to DNA damage response. Epigenetic regulation and spliceosome regulators have been thoroughly introduced, but the rest also need some paragraphs to introduce.
  4. Please provide more information and details about the CH in paragraph Line 300, like definition, consequences, and so on.
  5. Line 367, “5.1.” and Line 411, “5.1.”, please revise.
  6. Line 413, I cannot understand “19.958 individuals” in this sentence, is it “19,958”? According to the figure legend?
  7. Using the percentage value to present Figure 3 may be better,
  8. The authors introduced 5 main titles: 1. Mutations in MPN; 2 phenotypic mutations; 3. CH/MPN mutations in healthy individuals; 4. background population mutations; 5. CVD related. It seems that the paragraph “2.2. Algorithm integrating NGS data in patient prognosis” didn’t belong to any of them, authors may consider putting this paragraph as an independent one.
  9. Passenger mutations were not introduced in the manuscript, authors should mention that.

Reviewer 3 Report

The author performed a comprehensive overview of MPN genetic landscape, ranging from the first descriptions of the driver mutations to the more recent somatic mutations identification with NGS techniques, with a detailed discussion of the pathogenetic role of these mutations. Afterwards, the review focused on the significance of age-related acquisition of MPN-like somatic mutations and subsequent development of clonal hematopoiesis in the healthy individuals, both in terms of increased risk of cardiovascular events, and the possibility of a substantial under-diagnosis of MPNs, as suggested by a cross sectional study including 19.958 individuals, where the prevalence of JAK2V617F and CALR low mutant allele burdens was equal to 3.1% and 0.16%, respectively.

Comments: 

- The first report of CML by Bennet dates back to 1845 and not 1841. (Bennett, J.H. (1845) Case of hypertrophy of the spleen and liver in which death took place from suppuration of the blood. Edinburgh Medical and Surgical Journal, 64, 413±423).

- The "conclusions" section should be more concise, summarizing the "take home messages" of the topics extensively discussed above.
